# Evaluation of Second-Hand Exposure to Electronic Cigarette Vaping under a Real Scenario: Measurements of Ultrafine Particle Number Concentration and Size Distribution and Comparison with Traditional Tobacco Smoke

**DOI:** 10.3390/toxics7040059

**Published:** 2019-11-25

**Authors:** Jolanda Palmisani, Alessia Di Gilio, Laura Palmieri, Carmelo Abenavoli, Marco Famele, Rosa Draisci, Gianluigi de Gennaro

**Affiliations:** 1Department of Biology, University of Bari, Via Orabona 4, 70125 Bari, Italy; alessia.digilio@uniba.it (A.D.G.); lapalmieri@libero.it (L.P.); gianluigi.degennaro@uniba.it (G.d.G.); 2National Institute of Health, National Centre for Chemicals, Cosmetic products and Consumer Health Protection, Viale Regina Elena 299, 00161 Roma, Italy; carmelo.abenavoli@iss.it (C.A.); marco.famele@iss.it (M.F.); rosa.draisci@iss.it (R.D.)

**Keywords:** electronic cigarettes, ultrafine particles, size distribution, second-hand aerosol, tobacco smoke, indoor air quality

## Abstract

The present study aims to evaluate the impact of e-cig second-hand aerosol on indoor air quality in terms of ultrafine particles (UFPs) and potential inhalation exposure levels of passive bystanders. E-cig second-hand aerosol characteristics in terms of UFPs number concentration and size distribution exhaled by two volunteers vaping 15 different e-liquids inside a 49 m^3^ room and comparison with tobacco smoke are discussed. High temporal resolution measurements were performed under natural ventilation conditions to simulate a realistic exposure scenario. Results showed a systematic increase in UFPs number concentration (part cm^−3^) related to a 20-min vaping session (from 6.56 × 10^3^ to 4.01 × 10^4^ part cm^−3^), although this was one up to two order of magnitude lower than that produced by one tobacco cigarette consumption (from 1.12 × 10^5^ to 1.46 × 10^5^ part cm^−3^). E-cig second-hand aerosol size distribution exhibits a bimodal behavior with modes at 10.8 and 29.4 nm in contrast with the unimodal typical size distribution of tobacco smoke with peak mode at 100 nm. In the size range 6–26 nm, particles concentration in e-cig second-hand aerosol were from 2- (Dp = 25.5 nm) to 3800-fold (Dp = 9.31 nm) higher than in tobacco smoke highlighting that particles exhaled by users and potentially inhaled by bystanders are nano-sized with high penetration capacity into human airways.

## 1. Introduction

The electronic cigarette (e-cig) market has grown enormously in EU member states over the last years due to intense and strategic marketing campaigns made by producers and retailers who have advertised it as an aid to reducing and/or eliminating addiction to tobacco cigarette smoke [1,2]. Key strength of the marketing campaign has been also advertising e-cig use as a cheap and safer way to smoke vaporized chemicals such as nicotine in public places where smoking is banned and as a means of reducing the damage caused by passive smoking. E-cigs have been in substance marketed as ‘a new way of smoking’, technologically advanced, fashionable and healthier than conventional tobacco smoke. The public acceptance of e-cigs has been promoted by the mechanism of aerosol formation, perceived as healthier than traditional tobacco products combustion. Indeed, it is assumed that e-cigs do not generate harmful substances because, unlike traditional cigarettes, which burn tobacco leafs, the e-cigs vaporize liquid components at a lower temperature. In order to explore European attitude to e-cigs use, as a replacement or in conjunction with tobacco products, public opinion polls have been carried out over the years. The latest and more comprehensive survey was published in 2015 and carried out by TNS Opinion & Social network, on behalf of European Commission, in the 28 Member States, with the aim to explore the use of e-cigs among European citizens and to highlight the factors influencing the choice and diffusion of this device [3]. On the basis of collected data through face-to-face interviews, survey data highlighted an increase in the use of e-cigs between 2011–2014 and showed that e-cigs are mostly perceived as attractive by teenagers and young adults. Unfortunately, the effectiveness of e-cigs in stopping and reducing smoking could prove to be merely a marketing strategy due to inconsistencies of available literature data and the risk of a potential concomitant use of e-cigs and traditional cigarettes [4,5,6,7]. E-cigs are composed by a mouthpiece, an atomizer provided with a heating element, a tank suited to be filled with the liquid formulation, a manual switch and a re-chargeable battery. The rapid widespread of e-cigs stimulated industry to produce and place on the market more and more technologically advanced versions. However, they all work on the same operating principle: when the user inhales directly from the mouthpiece or presses the switch button, the electric heater is activated and the liquid formulation contained in the tank is vaporized in a fine mist of liquid droplets. In addition to many e-cigs typologies on the market, a wide range of liquids (e-liquids) with different composition and flavour exist. Smokers generally inhale a mixture of propylene glycol and glycerol (both used as food additives and solvents in pharmaceutics and cosmetics), various flavourings (most of them used as food additives), nicotine (in variable quantities or absent altogether) and, in smaller quantities, water. The widespread use of e-cigs attracted significant attention from governmental bodies, scientists and the medical profession in relation to their possible effects on public health. One much debated issue in the scientific community and among International Organizations concerns the potential risks due to short- and long-term inhalation exposure to chemicals present in e-liquids as contaminants/impurities (i.e., aromatic hydrocarbons, heavy metals and aldehydes) and e-liquid main components (i.e., propylene glycol and glycerol), taking into account that, in the latter case, long-term effects by direct inhalation in the lung cannot be predicted even though their use as food additives is approved [8]. Finally, but not less important, concerns have arisen in relation to potential risks due to active and passive exposure (e.g., main stream and second-hand stream vaping) to ultrafine particles. With the aim to elucidate all the debated issues, intensive scientific research has been carried out and several papers have been published in international journals on the evaluation of the potential impact of e-cig consumption on human health. Attention has been particularly paid on the determination of the chemical composition of e-liquids and the evaluation of particles formation. Several recently published papers pointed out the presence of harmful compounds in e-liquids and generated mainstream aerosols, even at lower levels than for tobacco cigarettes, such as Volatile Organic compounds (VOCs), Polycyclic Aromatic Hydrocarbons (PAHs), heavy metals and tobacco minor alkaloids. These investigations were performed directly on the formulas or on the mainstream aerosol by carrying out tailored experiments with smoking machines [9,10,11,12,13]. Compounds of concern detectable in the e-cig mainstream aerosol may be present in the liquid formulation as contaminants or may be generated by the vaporization process with concentration levels increasing with the increase in battery power [14]. The attention of scientists has been also paid on second-hand exposure, also called second-hand vaping (SHV), which occurs when e-cig aerosol is exhaled by users in public places or enclosed private environments. The purpose is to evaluate the extent of the impact of e-cig exhaled aerosol on indoor air quality and the potential passive exposure of bystanders in the environment where e-cig consumption occurs [15,16]. More specifically, passive exposure to e-cig aerosol in terms of fine and ultrafine particles was evaluated by carrying out investigations on particles number concentration and size distribution both in test emission chambers [17] and inside controlled real-scale rooms (i.e., meeting room, laboratory room, office, patient room, vape shop) with an internal volume ranging from 30 to 137.2 m^3^, enrolling volunteering vapers with the purpose to simulate as closely as possible the real exposure scenario and estimate the dose received by passive bystanders [9,18,19,20,21,22,23,24,25,26]. Over the years, the experimental design was more and more improved, involving more individuals, standardizing the experimental procedure, taking into account all the affecting parameters and evaluating both spatial and temporal distribution of particles number concentration. In some of the abovementioned studies, the comparison with the traditional tobacco cigarette smoke was also carried out [21,22]. To mention the most relevant findings recently published, Scungio et al. measured particle number concentration and size distribution of the exhaled aerosol from a single user related to 10 min vaping sessions inside a 40 m^3^ room. Particle number concentration in the range 6.30–9.08 × 10^3^ part/cm^3^ was measured and bimodal distribution with peaks at 30 and 90 nm was observed [24]. Zhao et al. enrolled 13 experienced e-cig users and asked them to perform vaping sessions of 10 min inside 80 m^3^ room under intensive ventilation (high air exchange rate, 4.1 h^−1^). Average particle number concentration during puffing was 2.48 × 10^4^ part/cm^3^ and size distribution showed two modes at about 15 and 85 nm [26]. In this context, the present work is placed. In reaction to the rapid spread of e-cigs on the Italian market as well as the lack of harmonized regulatory framework at European level, in 2013, the Italian Ministry of Health requested the first interventions to safeguard public health [27]. The National Institute of Health was empowered at that time to define research activities involving Italian research institutions, universities and control bodies in a joint work to evaluate potential risks related to e-cig consumption. As a result, a National research project entitled “New articles and new health risks: the electronic cigarette” was developed (2013–2016), with the priority objective, among the others, to evaluate the second-hand exposure due to e-cig use through the simulation of a realistic exposure scenario. In the framework of the aforementioned project relevant scientific data on the composition of e-liquids and generated aerosol were collected contributing to the harmonization process of the legislative frameworks of EU Member States, resulting in the implementation of the European Directive 2014/40/EU concerning manufacture, labeling and advertising of e-cigs [28]. Starting from the encouraging preliminary results obtained in the framework of the National project, an ad-hoc experimental campaign was carried out by the authors in 2017 in order to collect a larger set of reliable data on second-hand vaping exposure due to e-cig consumption in a real setting. Therefore, in the present paper, the characteristics of e-cig aerosol, in terms of ultrafine particles (UFPs) number concentration and size distribution, exhaled by two trained volunteering subjects vaping 15 different e-liquids inside a 49 m^3^ room according to a standardized experimental procedure, under controlled micro-environmental parameters and natural ventilation conditions, are discussed. For comparison, experiments with a popular brand of tobacco cigarette were carried out and conclusions on the relative impact of the second-hand vaping and smoking on bystanders were drawn.

## 2. Materials and Methods

### 2.1. Materials

In order to have a general view and representation of the e-cig market at European level, the National Institute of Health carried out a preliminary survey identifying the most popular brands. Fifteen refill liquids of five different brands with and without nicotine and characterized by different flavours were selected and purchased on-line from EU manufacturers or importers in 10–30 mL plastic bottles. Before being used for the experimental activity, e-liquids were properly stored and kept at room temperature and away from the direct sunlight as recommended on the product labels. More specifically, the selected refill liquids were manufactured in Italy (*n* = 10), China (*n* = 3), France (*n* = 1) and the United Kingdom (UK, *n* = 1). The general composition, as reported on products’ label, was propylene glycol, glycerol, flavours, nicotine and water. E-liquids composition in terms of propylene glycol and glycerol content (expressed in %), characteristic flavour and nicotine content (expressed as mg/mL or mg/g), as well as the country of manufacture is reported in Table 1. E-liquids are listed in the Table with progressive ID numbers that will be cited later in the text making the discussion on the obtained results easier (see Results and Discussion section). For e-liquids with ID 01, 02, 03, 08 and 10, belonging to the same Italian manufacture company, only qualitative composition is reported in Table 1 due to missing information on the percentage of single components on the label. E-cigs used for the experimental activity were second-generation refillable devices produced in China and were purchased from a popular Italian on-line store. This typology of device was button-activated and equipped with a cartomizer suited to be filled with the e-liquid, with a capacity of 1.6 mL. The cartomizers used in this experimental activity were disposable therefore a new cartomizer was used for each experiment in order to avoid contamination among e-liquids with different composition. The e-cig battery was not a variable voltage type; voltage value was fixed and equal to 4.2 V (420 mAh). E-cig resistance was 2.4 Ω. A popular brand of traditional cigarettes (Merit 100′s) with Nicotine content equal to 0.6 mg was chosen to compare contribution from e-cig to indoor particles concentration and size distribution with that from a conventional tobacco cigarette.

### 2.2. Experimental Design for Aerosol Characterization

Ad-hoc experimental design was developed in order to simulate as closely as possible a realistic second-hand vaping exposure scenario under the typical ventilation rate of naturally ventilated buildings according to d’Ambrosio et al. [29]. The simulation of second-hand vaping exposure scenario under natural ventilation conditions, therefore, reduced air exchange rate, represents an attempt to simulate a more realistic inhalation exposure scenario, likely to be found in private dwellings and in public places where, still today, recommended high air quality standards are not fully respected. More specifically, UFPs number concentration and size distribution in the aerosol generated from e-cig consumption were evaluated carrying out experiments inside a 49 m^3^ room provided with one window and one door under natural ventilation conditions. The room was furnished with one table and one chair, both placed in the middle. Moreover, it was not equipped with a mechanical air exchange and ventilation system (e.g., Heating Ventilation and Air Conditioning system, HVAC), but fans were placed inside the room to ensure that during the experiments, the air was adequately mixed. The air exchange rate (AER) inside the room was derived by means of a CO_2_ decay test [30], performed with a CO_2_ sensor (LSI Lastem, Stetta, Italy) monitoring gas concentration inside the room before, during and after the burning of an incense stick for 10 min. AER was determined under the same ventilation conditions adopted in the vaping/smoking tests, i.e., door and window closed, and the average obtained value was 0.1 h^−1^. Experiments were carried out under monitored environmental parameters, i.e., temperature 26 ± 2 °C and relative humidity 50 ± 5%, using a portable monitoring weather station (PCE-FWS 20 Weather Station, Easy weather pc software, PCE Instruments, PCE Holding, Meschede, Germany). Two volunteering vapers were recruited (vaper A, vaper B), and before the experimental campaign, they were asked to be involved in a training period regularly vaping e-cig for one month, approximately one hour a day. The volunteering vapers A and B were two adult females, 41 and 38 years old, respectively. The volunteers were mainly selected on the basis of their different attitude to vape/smoke as one was a former smoker of tobacco cigarettes (that had quit smoking few years before the experimental campaign) and the other one had never smoked in her life. In this way, although both the volunteers were involved in the same training period with e-cig, inter-vaper variability based on a different way of vaping/smoking was evaluated. For the experimental activity, the vapers were invited to perform for each selected refill a vaping session lasting 20 min, considered representative of the average time a common vaper uses e-cig. The e-cig was filled with the selected e-liquid outside of the room and given to the vaper before starting the vaping session. A fully charged battery was used before each experiment to avoid power drain. Only the vaper was allowed to stay inside the room during the background and the vaping sessions, sitting on a chair placed in the middle of the room. The window and the door were kept closed for the entire duration of each experiment. The experimental procedure was standardized in terms of the overall experiment duration and e-cig user topography (i.e., puff length, inter-puff interval and number of puffs per vaping session) as well as preliminary evaluation of the room background. The number of puffs for each vaping session was 40 and the puff duration and inter-puff interval were 3 s and 30 s, respectively. Puff duration of 3 s and puff intervals of 30 s are considered the standard conditions for aerosol generation as reported in Coresta Recommended Method (CRM) No. 81 for aerosol generation and collection [31,32]. The puff duration of 3 s is considered the necessary timing for the e-cig to reach a stable temperature [17,25]. Intra-vaper variability, and therefore, measurement repeatability, was evaluated performing experiments in triplicate for two specific e-liquids with nicotine (18 mg/g) and without nicotine and for the selected tobacco cigarette, with both the vapers. High temporal resolution number concentration and size distribution measurements of ultrafine particles, before, during and after the vaping/smoking sessions, were performed using the Fast Mobility Particle Sizer (FMPS 3091, TSI Incorporated, Shoreview, MN, USA) able to perform particles counting and classification on the basis of the electrical mobility technique [33,34]. FMPS sampling flow rate was 10 L/min and the particle size measurements range was from 5.6 to 560 nm. High flow rate operation mode ensures that both losses and evaporation of volatile and semivolatile particles are minimized. Measurements were performed with 1-s time resolution. The instrumentation was placed at 1 m with respect to the vaper’s position and the sampling inlet approximately at the same height as the vaper’s mouth. No different distances with respect to the vaper’s position were taken into account in this study; therefore, the spatial distribution was not investigated, and only the temporal variation of particles number concentration and size distribution was observed and discussed. Data collected for 20 min vaping sessions were compared with those related to one tobacco cigarette smoking session. The comparison was made taking into account that an average tobacco smoker will in 20 min generally smoke one cigarette (the overall consumption of tobacco cigarettes for an average smoker is 20 cigarettes/day). Therefore, the purpose was to simulate two exposure scenarios determined by vaping and smoking sessions representative of the average vapers/smokers behavior. The procedure for all tobacco cigarette experiments was also standardized (i.e., 1 puff every 30 s). A total of 44 experiments were carried out, 22 experiments (vaping and smoking sessions including replicates) for each volunteer. Each experiment was started when the particles number concentration in the room reached approximately the background level.

### 2.3. Ethical Statement

Volunteering vapers gave their informed consent for inclusion in the experimental activity signing a form properly prepared by the Ethical Committee of the National Institute of Health (project code: CCM 2013, date of approval: 12 December 2013). The study was conducted in accordance with the Declaration of Helsinki.

## 3. Results and Discussion

### 3.1. E-Cig Second-Hand Aerosol: Ultrafine Particle Number Concentration, Intra- and Inter-Vaper Variability

High temporal resolution monitoring inside the 49 m^3^ room before, during and after vaping sessions with each selected e-liquid for both the enrolled vapers allowed to observe a systematic increase in UFPs concentration (part cm^3^) related to the vaping activity. The background UFPs concentration levels, the peak values reached immediately after the end of the vaping session and the related increments (e.g., increase in UFPs concentration over the background, reported with Δ in the Table), for all 38 e-cig experiments, are reported in Table 2. Replicated experiments for e-liquids with ID 13 and 14 are marked with asterisk. UFPs concentration increments for the vaper A vary from the minimum value 1.37 × 10^4^ part cm^−3^ to the maximum value 4.01 × 10^4^ part cm^−3^, while for the vaper B from the minimum value 6.56 × 10^3^ part cm^3^ to the maximum value 3.28 × 10^4^ part cm^−3^. Differences observed in this study between the two vapers in terms of UFPs concentration increments suggest that, although the experimental procedure was standardized and all the affecting parameters (i.e., number of puffs per vaping session, vaping session duration, puffing time, interval between puffs) were kept constant in all the performed experiments, the inhalation volume and the depth of inhalation during the vaping activity may be different between two subjects and represent a variability factor. Therefore the individual vaping mode in terms of inhalation volume and depth of inhalation may result in a significant difference in the user’s exhaled aerosol and in particles number concentration level in the room air (inter-vaper variability). The effect of vaping mode on particles formation in terms of number concentration has been already investigated but, to the authors knowledge, in the mainstream aerosol generated by smoking machines [35,36]. Fuoco et al. demonstrated the positive correlation between the puffing time and the total number concentration of generated particles, with all the other parameters affecting on particles formation kept constant. More specifically, the longer the puffing time, the higher the particle number concentration was in the mainstream aerosol, likely due to increased performances of the battery allowing to further promote e-liquid evaporation process [35]. Mikheev et al. showed that the concentration of particles increased when the puffing flow rate was increased from 15 to 45 mL/min and that flow rate effect was more evident for nano-sized particles in the size range 5–40 nm [36]. Anyway, although these investigations were useful to deepen e-cig aerosol formation and identify the extent of the influence of single parameters, information they provided were related to mechanical systems operating in a repeatable way. At a fixed puffing time and puffing flow rate, the smoking machine systematically determines the e-cig aerosol formation in the same manner. On the contrary, e-cig aerosol formation due to vaping activity from an individual human subject and among different subjects is affected by a higher level of complexity. Given a fixed puffing time (3 s in this study), the subject behavior in vaping may be different because the inhalation volume (related to the puffing flow rate) and the exhaled aerosol may differ from one puff to another one made by the same subject (intra-subject variability); moreover, from one subject to another one (inter-subjects variability). The data collected for e-cigs experiments carried out in this work highlight in a remarkable way the inter-vaper variability. Another purpose of the study was to investigate if the nicotine content of e-liquids could have a remarkable effect on aerosol formation and particle concentration the bystanders are exposed to and if this effect could be observed with the proposed experimental design. To this purpose, UFPs concentration increments due to the consumption of e-liquids couple with ID 01 and 08, 05 and 06, 07 and 11 characterized by the same flavour (mint, tobacco and anise respectively) and basic composition (same percentage of propylene glycol and glycerol) and manufacture company but with different nicotine contents (0 and 18 mg/mL) were compared (Table 2). UFPs number concentration measurements for the experiments with the abovementioned e-liquids does not allow to argue, as done in previous investigations [22,24], that given the e-liquid formulation, the nicotine content has a direct effect on aerosol formation and as a result on UFPs concentration exhaled by the user and measured inside the room. The obtained results in this study do not lead to an unambiguous conclusion. It is possible to observe, indeed, that only in one case, the use of nicotine containing e-liquid may be associated with a higher measured UFPs concentration increment inside the room. More specifically, for the experiment involving the vaper A, UFPs concentration increment was equal to 1.43 × 10^4^ part cm^−3^ for free Nicotine e-liquid (ID 01) and 3.34 × 10^4^ part cm^−3^ for nicotine containing e-liquid (ID 08). In all the other experiments involving both the vapers and the selected e-liquids for comparison, the observed UFPs concentration increment during the vaping session for Nicotine free and Nicotine containing e-liquids was comparable (experiments with ID 07/11 for both the vapers and with ID 05/06 for vaper B) or slightly higher in the case of Nicotine free e-liquid (experiment with ID 05/06 for vaper A and 01/08 for vaper B). The intra-subject variability was also evaluated asking the vapers to perform replicated experiments (in triplicate) with two selected e-liquids (ID 13 and 14). Regarding the experiments performed by the vaper A, the mean value of UFPs concentration increment was (2.14 ± 0.27) × 10^4^ and (3.14 ± 0.35) × 10^4^ part cm^−3^ for e-liquid with ID 13 and 14, respectively. For comparison, the mean value of UFPs concentration increment derived by the experiments performed by the vaper B was (1.63 ± 0.28) × 10^4^ and (2.36 ± 0.10) × 10^4^ part cm^−3^ for e-liquid with ID 13 and 14, respectively. RSD% ranged from the minimum value of 4.2% (ID 14, vaper B) to the maximum value of 17.3% (ID 13, vaper B). Vaper B showed higher intra-subject variability compared with vaper A, whose RSD% associated to the two sets of replicated experiments was approximately the same (11.1 and 12.6%). Intra- and inter- subjects variability has been observed in other studies, e.g., Zhao et al. found large variation in mean particle number concentration values between puffs produced by an individual user and across 13 different users [26]. More specifically, large inter-subjects variability across the vapers was demonstrated by the mean value of mean particle number concentration per puff produced by each involved subject ranging from 8.05 × 10^3^ to 4.0 × 10^4^ part cm^−3^. 

### 3.2. E-cig Second-Hand Aerosol: Temporal Profiles of UFPs Number Concentration

Figure 1 shows the temporal trend for UFPs number concentration in the room derived through particles monitoring before, during and after e-cig vaping sessions for 12 h. The reported trends are related to two e-cig experiments with a selected e-liquid (ID 14, nicotine free) involving the vaper A and vaper B. The reported trends can be considered representative for all the performed e-cig experiments. Regarding the 20-min vaping event (indicated by start and stop in figure), an increase of one order of magnitude in particle number concentration for both the experiments was observed (from the background 3.94 × 10^3^ part cm^−3^ to the peak value 3.28 × 10^4^ part cm^−3^ for vaper A, from the background 2.86 × 10^3^ part cm^−3^ to the peak value 1.62 × 10^4^ part cm^−3^ for vaper B). The increase in UFPs concentration was more evident for vaper A (Δ: 2.89 × 10^4^ part cm^−3^) than for vaper B (Δ: 1.33 × 10^4^ part cm^−3^), highlighting the occurrence of inter-vaper variability, as already discussed in the previous subsection (Table 1). Both the reported trends show that, immediately following the vaping session at *t* = 20 min, particle number concentration decreases fast with a decay rate higher during the first two hours and lower over the remaining time. Particle concentration decay after the vaping event may be related to multiple and simultaneous processes occurring inside the room. To a greater extent, the decay is affected by the dilution and evaporation in the room volume, as e-cig particles are mainly composed by liquid droplets formed by supersaturated vapors of volatile compounds such as propylene glycol, glycerol and additives [9,17,26]. The evaporation process occurred at higher rate during the two hours immediately after the end of the vaping session likely due to the low ambient partial pressure of aerosol constituents in the room air. It is reasonable to assume that, as time went by, the ambient partial pressure of aerosol constituents in the room air increased getting closer to the vapor pressure and, as a result, the evaporation rate was reduced and the concentration decay was slower. To a minor extent the decay is related to removal mechanisms such as adsorption on surfaces, deposition and exfiltration through the room leaks (i.e., joints at window and cracks at door) [21].

### 3.3. E-Cig Second-Hand Aerosol: Particle Size Distribution and Temporal Evolution

For all e-cig experiments, the particle size distribution of second-hand aerosol was bimodal with one mode located at 10.8 nm and the other one at 29.4 nm, as shown in Figure 2a, where particle size distributions (dN/dLogDp, #/cm^3^), each averaged for all the experiments carried out for vaper A and vaper B, respectively, are reported. Therefore, it is evident from the investigation that most of the second-hand aerosol particles, exhaled by the user and potentially inhaled by the passive bystander, are nano-sized particles. It was already highlighted that the aerosol exhaled by a vaper’s mouth is subjected to evaporation and dilution in the indoor environment resulting in high concentration of smaller particles but it is also important to underline that the evaporation process of the liquid droplets starts in the vaper’s respiratory system leading to the exhalation of small-sized particles [17]. Similar findings on second-hand aerosol size distribution were reported in previously published papers [17,24,26]. For instance Scungio et al. highlighted that second-hand aerosol due to e-cig consumption, under similar experimental conditions of the present study, was characterized by a bimodal distribution, but, compared with our results, with a size shift of the mode values (main mode at 30 nm and secondary one at about 90 nm) [24]. Inhalation of nano-sized particles may represent a risk factor for the exposed bystanders, as it is well documented that the smaller the particle diameter, the higher the penetration in the deepest tracts of human respiratory apparatus is [37,38]. Nano-sized particles, due to size, may deposit efficiently by diffusion in all regions of respiratory tract and may serve as a preferential delivering system for chemicals, either formed during vaporization process or included in the e-liquid formulation as contaminants, at bronchiolar and alveolar levels where absorption in the blood stream occurs There is evidence that alveolar macrophages involved in removal mechanism of particles entered into human airways are not able to efficiently remove nano-sized particles (e.g., phagocytosis impairment), thus enabling them to deeply access into the lung and come into close contact with the alveolar epithelium [37]. Deepest the penetration longer the deposition time is, resulting in higher probability of adverse effects due to particle-tissue and particle-cell interaction [39]. More specifically, once reached the alveolar epithelium, nanoparticles are able to cross the blood-air tissue barrier and enter into the blood stream accessing to other target organs. Therefore, the assisted access allows chemicals soluble in nano-sized propylenglycol/glycerol particles produced during e-cig consumption to induce adverse effects, depending on chemicals toxicity potential. In the light of the above considerations, the particle size has to be considered the key factor to take into account for potential risk evaluation, as it will be underlined in the following subsection when a comparative discussion on e-cig aerosol and tobacco cigarette smoke is done. In Figure 2b the particle size distribution temporal evolution (dN/dLogDp, #/cm^3^), corresponding to the number concentration trend reported in Figure 1 (vaper A) is shown. Particle size distributions related to the background (t_0_), at the end of the vaping session (t_20 min_) and different hours after the end of the vaping event (t_1 h_, t_2 h_, t_3 h_...t_10 h_) are compared. The evolution was followed until 10 h after the end of the vaping session. It is possible to observe that, 1 h after the end of the vaping event (t_1 h_), the particle number concentration at 10.8 nm and 29.4 nm was reduced by 50% and 65%, respectively.

During the successive hours particle concentration at peak values continues to decrease slower and slower until size distributions become comparable. Moreover, particle size shift during the decay was not observed confirming that the aerosol dilution process inside the room is faster than the coagulation one coherently with what already observed by Avino et al. [18].

### 3.4. Comparison of E-Cig Second-Hand Aerosol with Environmental Tobacco Smoke 

In this subsection, considerations made on the comparison between e-cig second-hand aerosol and environmental tobacco smoke in terms of UFPs number concentration and size distribution are reported. In Table 3 background UFPs concentration levels, the peak values reached immediately after the end of the smoking session (at *t* = 7 min) and the related increments (reported as Δ in the Table) for six tobacco cigarette experiments (three replicates for each smoker) are reported. 

UFPs concentration increment ranged from 1.31 × 10^5^ to 1.46 × 10^5^ part cm^−3^ for smoker A and from 1.12 × 10^5^ to 1.32 × 10^5^ part cm^−3^ for smoker B. Therefore, mean UFPs concentration increment values were (1.38 ± 0.08) × 10^5^ part cm^−3^ and (1.23 ± 0.10) × 10^5^ part cm^−3^ for smoker A and B, respectively. Moreover, RSD% was 5.5% for smoker A and 8.1% for smoker B, suggesting, also for tobacco cigarette experiments, higher intra-smoker variability for the smoker B. Figure 3 shows UFPs concentration temporal profiles (#/cm^3^) due to tobacco cigarette use made by smokers A and B in a single experiment (one of the three replicated measurements, continuous lines). For comparison, UFPs concentration temporal profiles (#/cm^3^) for e-cig experiments with a selected e-liquid (ID 14) are also shown (dotted lines). The e-liquid ID 14 was selected for comparison as it determined, for each vaper, an average UFPs concentration increment (Table 2). 

It is evident that UFPs concentration measured inside the room at the end of the smoking event (peak value at *t* = 7 min) for both the volunteers and therefore UFPs concentration increments (Δ) are one up to two order of magnitude higher than those observed for e-cigs experiments. More specifically, UFPs concentration peak values and related increments for tobacco cigarette experiments were 1.47 × 10^5^ and 1.46 × 10^5^ part cm^−3^ for smoker A and 1.35 × 10^5^ and 1.32 × 10^5^ part cm^−3^ for smoker B, respectively. Peak values and increments with respect to the background values were approximately the same. For comparison, UFPs concentration peak values and related increments for e-cig experiments were significantly lower, in more detail 4.65 × 10^4^ and 3.61 × 10^4^ part cm^−3^ for smoker A and 1.35 × 10^4^ and 3.28 × 10^4^ for smoker B, respectively. Therefore, it is possible to state that the potential impact on indoor air quality determined in an enclosed environment with natural ventilation conditions (e.g., no mechanical air exchange/ventilation systems and door and window closed), and, as a result, the potential exposure of the passive bystanders in terms of UFPs number concentration is significantly higher in the case of a single tobacco cigarette consumption compared with 20-min e-cig vaping. 

This finding was expected, in agreement with previous studies focused on comparison between e-cig and tobacco cigarette [18,21,22], given the tobacco cigarette combustion process and the emission behavior, e.g., continuous particles emission during cigarette consumption resulting in both mainstream and sidestream smoke. Particle size distribution is, instead, the key issue when e-cig second-hand aerosol and environmental tobacco smoke are compared under the simulation of a realistic exposure scenario. In contrast to e-cigs, particle size distribution of the environmental tobacco smoke provides a single mode with a maximum roughly at particle diameter (Dp) 100 nm. This is shown in Figure 4a, where particle size distributions (dN/dLogDp, #/cm^3^), each averaged for all tobacco experiments carried out for smoker A and smoker B, respectively, are reported. Unimodal size distributions are peaked at 91.3 nm. Figure 4b allows the analysis of temporal evolution of particle size distribution for tobacco smoke (data of a single experiment from the end of the smoking event until 6 h after). The evolution profile highlights that in two hours the particle number concentration indoors (peak value) was reduced of 70% (from 2.37 × 10^5^ to 6.96 × 10^4^ part cm^−3^) and that, over time, particle size shift from 93.1 nm to 107.5 nm occurred, suggesting that, starting from the end of the smoking event, the particles released indoors are subjected to growth. These results are in agreement with those obtained by Schripp et al. and Avino et al. [17,18]. 

The concentration decay (Figure 3, continuous lines; Figure 4b) is expected to be mainly due to dilution into the room air and deposition process of particles and adsorption onto the available surfaces in the room (i.e., furnishings, smoker’s clothing, smoker’s skin). The successive release of the adsorbed particles long after cigarette is smoked would lead both the smoker [40] and passive bystanders to third-hand exposure but this issue is not addressed in the present paper. Finally, Figure 5 shows the comparison between particle size distributions for e-cig second-hand aerosol and environmental tobacco smoke produced in all the experiments performed in this study (averaged particle size distributions). 

It appears evident that UFPs concentration at a size equal to 100 nm produced by tobacco consumption was significantly higher than that determined during e-cig vaping sessions. On the contrary, it is possible to observe that UFPs concentration in the environmental tobacco smoke within the size range 6.04–25.5 nm is negligible if compared with that related to e-cig aerosol. More specifically, in the nano-sized range, UFPs concentration in the e-cig second-hand aerosol revealed to be from 2- (at Dp = 25.5 nm) to 3800-fold (at Dp = 9.31 nm) higher than in tobacco environmental smoke. The exposure to a high concentration of nano-sized particles is certainly exacerbated by low air exchange rates in the environments where e-cig vaping may occur, but is also promoted by the fact the e-cig second-hand aerosol inhalation by passive vapers is perceived as more acceptable than tobacco smoke due to pleasantness of the characteristic flavours. Therefore, although tobacco cigarette use indoors is recognized as one of the main sources of pollutants both in gaseous and particulate phase resulting in high exposure levels for both active and passive smokers to carcinogenic compounds and particulate matter, e-cig second-hand aerosol exposure is not risk-free due to high concentration of nano-sized particles whose penetration capacity and delivering action of chemicals/contaminants into the deepest regions of human respiratory system is an issue of concern for public health and especially for most vulnerable populations, i.e., children and elderly people. 

### 3.5. Limitations

Limitations of the present study may be mainly related to the limited number of both volunteering vapers and investigated e-liquids. Enrolling a higher number of volunteers in the experimental activity would have certainly allowed to investigate more deeply inter-vaper variability. Investigations were carried out using a limited number of e-liquids (15) that, although selected by the National Institute of Health as representative of the most popular brands manufactured and imported on EU market, may not be enough to describe the variability in terms of chemical composition of all e-liquids commercially available and, as a result, to fully describe potential correlations between chemical composition and UFPs number concentration produced and exhaled. Moreover, the experimental activity was carried with only one model of e-cig and differences in terms of UFPs production potentially related to e-cig battery power cannot be discussed. Finally, spatial distribution of particles once exhaled by the vapers, not investigated in the present study, would have allowed to determine the distance of bystanders from the e-cig user where the UFPs concentration increment was not observed. 

## 4. Conclusions

The present paper shows the results obtained by an experimental campaign aimed to evaluate second-hand exposure from e-cig vaping, in terms of ultrafine particles number concentration and size distribution, and to make a comparative study with tobacco cigarette smoke. Investigations were carried out in a 49 m^3^ room with natural ventilation conditions simulating an exposure scenario commonly found in private dwellings and public places. The experimental results obtained allowed the authors to point out that, inside an enclosed environment with natural ventilation conditions and low air exchange rate, the potential exposure of the passive bystanders in terms of UFPs number concentration is significantly higher in the case of a single tobacco cigarette consumption compared with 20-min e-cig vaping, regardless of the e-liquid used. However, this study highlighted that the passive exposure to e-cig second-hand aerosol is not risk-free. Although the comparison in terms of UFPs number concentration indicates a lower impact indoors of e-cig use with respect to tobacco cigarette consumption, the analysis of the particles size distributions reveals that e-cig second-hand aerosol, exhaled by the user and potentially inhaled by the passive bystanders, is mainly composed of nano-sized particles within the size range of 6–26 nm (particles concentration from 2- at Dp = 25.5 nm to 3800-fold at Dp = 9.31 nm higher than in tobacco environmental smoke). The inhalation exposure to high level concentrations of nano-sized particles may represent a high level risk factor for passive vapers due to particles high penetration capacity and delivering action of chemicals/contaminants in the deepest regions of the human respiratory apparatus. Therefore, this study underlines that the particle size has to be considered a key factor to take into account by experts in indoor air and risk assessment fields, when passive exposure risk evaluation due to both e-cig and tobacco cigarette use is required. 

## Figures and Tables

**Figure 1 toxics-07-00059-f001:**
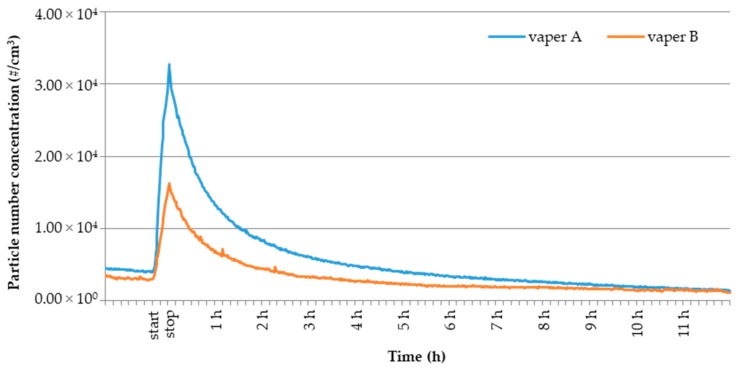
Temporal profiles of UFPs number concentration (#/cm^3^) related to experiments with e-liquid ID 14 performed by vaper A (blue line) and vaper B (red line). Start and stop define the vaping session.

**Figure 2 toxics-07-00059-f002:**
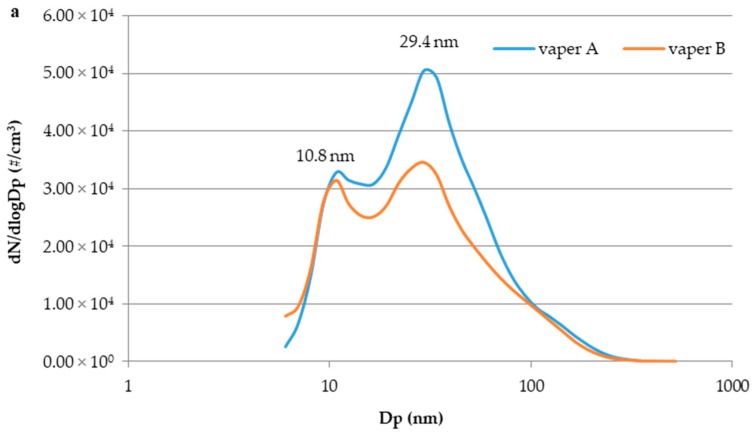
(**a**) Particle size distribution (dN/dLogDp, #/cm^3^) of second-hand aerosol generated by vaper A and vaper B (averaged for all the experiments); (**b**) Particle size distribution temporal evolution (dN/dLogDp, #/cm^3^) corresponding to particle number concentration trend reported in Figure 1 (vaper A).

**Figure 3 toxics-07-00059-f003:**
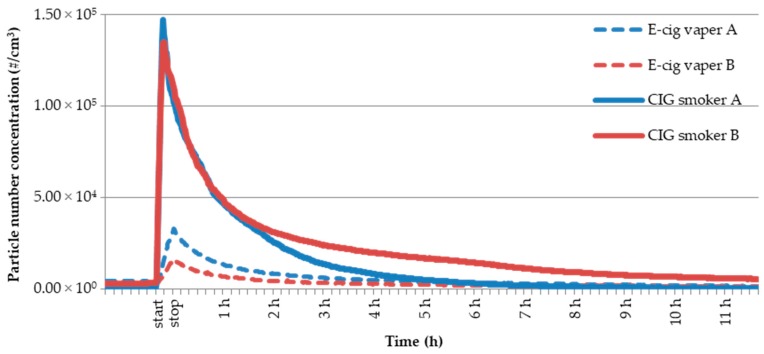
UFPs concentration temporal profiles (#/cm^3^) due to tobacco cigarette use made by smokers A and B in a single experiment (continuous lines). For comparison, UFPs concentration temporal profiles (#/cm^3^) for e-cig experiments with a selected e-liquid (ID 14) (dotted lines).

**Figure 4 toxics-07-00059-f004:**
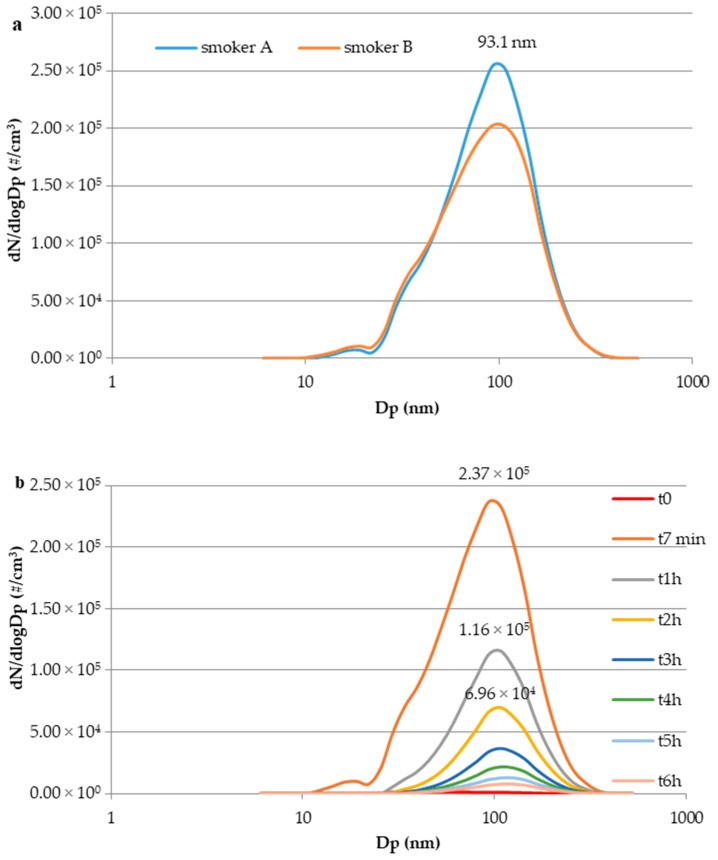
(**a**) Particle size distribution (dN/dLogDp, #/cm^3^) of environmental tobacco smoke generated by smoker A and smoker B (averaged for all the experiments); (**b**) particle size distribution temporal evolution (dN/dLogDp, #/cm^3^) of a single tobacco cigarette experiment.

**Figure 5 toxics-07-00059-f005:**
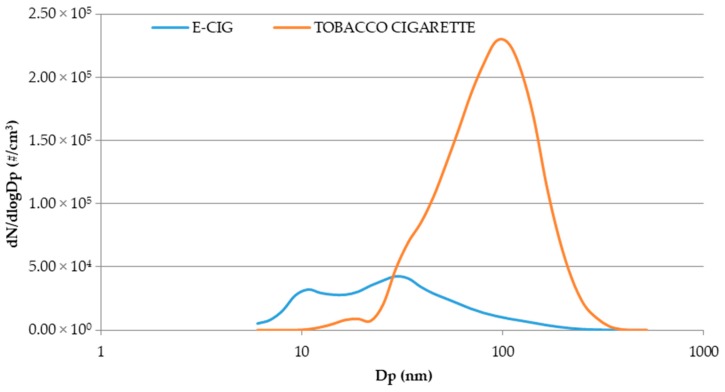
Comparison of particle size distributions of e-cig second-hand aerosol and environmental tobacco smoke. Size distributions are averaged for all the experiments and for both the vapers/smokers.

**Table 1 toxics-07-00059-t001:** E-liquids composition and information: percentage of the main components, nicotine content (mg/mL or mg/g), characteristic flavour and country of manufacture.

E-Liquid ID	Propylen Glycol (%)	Glycerol (%)	Water (%)	Flavour	Nicotine (mg/mL, * mg/g)	Manufacture (Country)
01	not declared	not declared	not declared	Mint	0	Italy
02	not declared	not declared	not declared	Cuban cigar	0	Italy
03	not declared	not declared	not declared	Rhum	0	Italy
04	50	40	5–10	Biscuit	0	Italy
05	50	40	5–10	Tobacco USA	0	Italy
06	50	40	5–10	Tobacco USA	18	Italy
07	50	40	5–10	Anise	0	Italy
08	not declared	not declared	not declared	Mint	18	Italy
09	75	25	/	Almond	11 *	China
10	not declared	not declared	not declared	Liquirice	18	Italy
11	50	40	5–10	Anise	18	Italy
12	75	25	/	Virginia tobacco	18 *	China
13	75	25	/	Cigar	18 *	China
14	>80	<20	not declared	Habanos Cigar	0	France
15	>80	<20	not declared	Cuban cigar	0	UK

**Table 2 toxics-07-00059-t002:** UFPs formation during e-cig experiments vaping sessions with vapers A and B: particles number concentration (#/cm^3^) at start and stop of the vaping sessions and related increment (Δ).

E-Liquid ID	Vaper A			Vaper B		
	UFPs Concentration Vaping Start (#/cm^3^)	UFPs Concentration Vaping Stop (#/cm^3^)	Δ (#/cm^3^)	UFPs Concentration Vaping Start (#/cm^3^)	UFPs Concentration Vaping Stop (#/cm^3^)	Δ (#/cm^3^)
01	7.24 × 10^3^	2.15 × 10^4^	1.43 × 10^4^	1.21 × 10^4^	3.71 × 10^4^	2.50 × 10^4^
02	8.99 × 10^3^	2.27 × 10^4^	1.37 × 10^4^	9.04 × 10^3^	1.56 × 10^4^	6.56 × 10^3^
03	8.28 × 10^3^	3.18 × 10^4^	2.35 × 10^4^	8.33 × 10^3^	1.56 × 10^4^	7.27 × 10^3^
04	1.04 × 10^4^	4.65 × 10^4^	3.61 × 10^4^	8.77 × 10^3^	4.16 × 10^4^	3.28 × 10^4^
05	1.49 × 10^4^	5.50 × 10^4^	4.01 × 10^4^	7.97 × 10^3^	2.41 × 10^4^	1.61 × 10^4^
06	1.22 × 10^4^	4.94 × 10^4^	3.72 × 10^4^	9.55 × 10^3^	2.69 × 10^4^	1.74 × 10^4^
07	1.14 × 10^4^	3.91 × 10^4^	2.77 × 10^4^	1.13 × 10^4^	2.70 × 10^4^	1.57 × 10^4^
08	9.24 × 10^3^	4.26 × 10^4^	3.34 × 10^4^	1.15 × 10^4^	2.43 × 10^4^	1.28 × 10^4^
09	7.41 × 10^3^	3.82 × 10^4^	3.08 × 10^4^	9.07 × 10^3^	2.72 × 10^4^	1.81 × 10^4^
10	1.13 × 10^4^	4.13 × 10^4^	3.00 × 10^4^	1.01 × 10^4^	3.75 × 10^4^	2.74 × 10^4^
11	8.53 × 10^3^	3.50 × 10^4^	2.65 × 10^4^	1.08 × 10^4^	2.54 × 10^4^	1.46 × 10^4^
12	1.09 × 10^4^	3.58 × 10^4^	2.49 × 10^4^	1.28 × 10^4^	2.98 × 10^4^	1.70 × 10^4^
13	7.11 × 10^3^	3.16 × 10^4^	2.45 × 10^4^	1.02 × 10^4^	2.68 × 10^4^	1.66 × 10^4^
13 (2) *	8.35 × 10^3^	2.88 × 10^4^	2.05 × 10^4^	1.15 × 10^4^	3.05 × 10^4^	1.90 × 10^4^
13 (3) *	8.03 × 10^3^	2.74 × 10^4^	1.94 × 10^4^	1.60 × 10^4^	2.94 × 10^4^	1.34 × 10^4^
14	8.87 × 10^3^	3.72 × 10^4^	2.83 × 10^4^	1.38 × 10^4^	3.85 × 10^4^	2.47 × 10^4^
14 (2) *	1.12 × 10^4^	4.37 × 10^4^	3.25 × 10^4^	1.42 × 10^4^	3.69 × 10^4^	2.27 × 10^4^
14 (3) *	1.17 × 10^4^	4.75 × 10^4^	3.58 × 10^4^	1.41 × 10^4^	3.76 × 10^4^	2.35 × 10^4^
15	1.25 × 10^4^	5.11 × 10^4^	3.86 × 10^4^	1.14 × 10^4^	3.49 × 10^4^	2.35 × 10^4^

* replicated measurements.

**Table 3 toxics-07-00059-t003:** UFPs formation during tobacco cigarette experiments with smokers A and B: UFPs number concentration (#/cm^3^) at start and stop of the smoking session and related increment (Δ).

Tobacco Exp.	Smoker A			Smoker B		
	UFPs Concentration Smoking Start (#/cm^3^)	UFPs Concentration Smoking Stop (#/cm^3^)	Δ (#/cm^3^)	UFPs Concentration Smoking Start (#/cm^3^)	UFPs Concentration Smoking Stop (#/cm^3^)	Δ (#/cm^3^)
exp. 1	9.43 × 10^2^	1.47 × 10^5^	1.46 × 10^5^	3.81 × 10^3^	1.16 × 10^5^	1.12 × 10^5^
exp. 2	2.10 × 10^3^	1.33 × 10^5^	1.31 × 10^5^	5.74 × 10^3^	1.31 × 10^5^	1.25 × 10^5^
exp. 3	2.70 × 10^3^	1.41 × 10^5^	1.38 × 10^5^	3.31 × 10^3^	1.35 × 10^5^	1.32 × 10^5^

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
