# Peer review of "Evaluation of Second-Hand Exposure to Electronic Cigarette Vaping under a Real Scenario: Measurements of Ultrafine Particle Number Concentration and Size Distribution and Comparison with Traditional Tobacco Smoke"

_toxics, 2019, doi:10.3390/toxics7040059_

Round 1

Reviewer 1 Report

The work is clearly written and presented and addresses a matter of growing concern. However, there is already a substantial body of literature on the matter. Authors need to more clearly state how they are advancing the current state of science and differentiate their novelty from the many works cited by the authors and others on the matter. I recommend addition of a table where literature on the matter is summarized (study, ventilation setting, e-cig characteristics, sample size, PNC values, mode diameters, etc.) which would also be helpful for readers on the special issue on the matter.

Please introduce paragraph breaks in the Introduction and Results section 3.1. It is very difficult to read it in the current form of two page long solid text.

Perhaps, proof read for grammar once. There are some minor errors.

Reviewer 2 Report

This is an important experimental contribution to second-hand exposure and particle size distribution of aerosols in a 49 m3 room from second-generation refillable e-cigarettes with disposable cartomizers at 4.2 V (420 mAh) and resistance of 2.4 Ω, compared to one brand of traditional cigarettes. Limitations of the study are missing. The paper could be shortened by eliminations of repetitions. In the introduction results of other studies are given in detail, which should rather be placed in the discussion in comparison with own results, explaining discrepancies. Line 183: Was "vaper A" or "vaper B" a never-smoker? Could this explain some differences between A and B? Possible carrier effects of nano-sized particles into alveoli, lung interstitium and blood stream could be discussed.
